# Zoonotic Risk of *Encephalitozoon cuniculi* in Animal-Assisted Interventions: Laboratory Strategies for the Diagnosis of Infections in Humans and Animals

**DOI:** 10.3390/ijerph18179333

**Published:** 2021-09-03

**Authors:** Antonio Santaniello, Ilaria Cimmino, Ludovico Dipineto, Ayewa Lawoe Agognon, Francesco Beguinot, Pietro Formisano, Alessandro Fioretti, Lucia Francesca Menna, Francesco Oriente

**Affiliations:** 1Department of Veterinary Medicine and Animal Productions, Federico II University of Naples, 80134 Naples, Italy; ludovico.dipineto@unina.it (L.D.); alessandro.fioretti@unina.it (A.F.); 2Department of Translational Medical Sciences, Federico II University of Naples, 80131 Naples, Italy; ilariacimmino@hotmail.it (I.C.); ayewa.agognon@gmail.com (A.L.A.); beguino@unina.it (F.B.); fpietro@unina.it (P.F.); foriente@unina.it (F.O.)

**Keywords:** zoonosis, *E. cuniculi*, laboratory diagnosis, *Microsporidium*, animal assisted interventions (AAIs), one health, patient

## Abstract

The involvement of animals for therapeutic purposes has very ancient roots. To date, it is clear that animal-assisted interventions (AAIs), in addition to ensuring the replacement of missing or deficient affects, improves psychophysiological parameters connected to human health. However, AAI could potentially present risks related to the transmission of infectious agents from animals to humans. Among these microorganisms, *E. cuniculi* is a microspore which induces pathological effects (fever, headache, nausea, vomiting, diarrhea, breathlessness, respiratory symptoms, and weakness) in both humans and animals. Consequently, an accurate and fast diagnosis of *E. cuniculi* infection, as well as the identification of new diagnostic approaches, is of fundamental importance. This literature review was carried out to provide an extensive and comprehensive analysis of the most recent diagnostic techniques to prevent and care for *E. cuniculi*-associated risks in the AAI field.

## 1. Introduction

*Encephalitozoon cuniculi* is a microsporidial, unicellular, spore-forming, obligate intracellular parasite and may infect different mammalian hosts, such as lagomorph, rodents, domestic carnivores, ruminants, and other animal species including non-human primates and humans [1,2,3]. This microsporidium represents an opportunistic pathogen in human patients with acquired immunodeficiency syndrome and other immunocompromised people [4,5,6,7]. Notably, *E. cuniculi* infections were reported also in immunocompetent people [8,9,10]. In the past, the phylogenetic origin of microsporidia has been an important matter of argument because they have many particular characteristics with respect to other microorganisms. Microsporidia were originally considered to be very basic eukaryotes due to their lack of mitochondria. Nevertheless, highly reduced and small mitochondria, named mitosomes, have been identified at the level of the cell organelle and are now considered to be an important distinctive feature of microsporidia. The close correlation between microsporidia and atypical fungi with extreme host cell dependency is supported by the discovery of a gene for this mitochondrial-type chaperone, combined with phylogenetic analysis of multiple gene sequences [11]. Furthermore, microsporidial spores possess fungal elements comprising fungal proteins, such as tubulins, trehalose, and chitin [12].

Microsporidia survive only by living in other cells and have the ability to produce environmentally resistant spores. Host cell infection occurs through the involvement of a specialized extrusion apparatus recognized as the polar tube or filament [11].

In this regard, it is worth noting that animals represent not only a potential source of zoonosis but also a great resource for the patients who relate to them [13] as part of a therapeutic intervention program [14]. The scientific literature in this area is very wide and reports a series of evidence concerning mainly the benefits that the animal brings to people [15], but it is rather scarce regarding potentially transmitted zoonotic pathogens.

Therefore, this literature review was conducted to highlight the importance of laboratory strategies for the diagnosis of *E. cuniculi* to stem the potential zoonotic risk of this microsporidium for people involved in animal assisted interventions (AAIs). Indeed, interaction between animals and humans has been associated with a lower risk of developing diseases of the cardiovascular system as a consequence of lower stress hormone concentrations (e.g., cortisol) and the increased release of neurotransmitters capable of facilitating social relationships (e.g., oxytocin) [16]. So far, *E. cuniculi* diagnosis relies on different serological and molecular techniques with variable highlights and challenges that will be further elucidated. The emerging information could be of valid help for health workers who, with different professional skills and in the context of One Health, work in the context of AAIs, in which the animals involved should always be subjected to specific health checks, even for often underestimated pathogens.

## 2. Materials and Methods

For our paper, a scientific literature search was performed through 30 June 2021 examining the National Library of Medicine “PubMed.gov” (accessed on 30 July 2021) based on the words “Encephalitozoon cuniculi” and “rabbit” or “rodents” or “dog” or “cat” or “horse” or “donkey” or “human”: “animal-assisted interventions”, “zoonosis” or “zoonoses” or “zoonotic risk” and “diagnosis”. The authors selected articles describing epidemiology, zoonotic potential, and related diagnostic techniques for *E. cuniculi*. Animal species were chosen based on the relevance of scientific literature. The titles of all obtained articles were screened for relevance to the topic, and, subsequently, consistent titles were assessed and selected based on their related abstracts. Additional studies were found after consulting the bibliographies of the screened papers. Only original papers in English were included.

## 3. Encephalitozoon Cuniculi

### 3.1. Infection in Animals

#### 3.1.1. Rabbits

Encephalitozoonosis has been described worldwide in laboratory, pet, and farm rabbits. Particularly, based on the serological evidence, wild rabbits (*Oryctolagus cuniculus*) represent the natural host of *E. cuniculi* [17].

In this microsporidium species, four genotypes (I, II, III, and IV) are recognised based on the number of repeats of 4-base sequences (5′-GTTT-3′) in the internal transcribed spacer region of the ribosomal RNA gene [18,19,20]. Based on the host species of the originally characterized isolates, these strains were also recognized as “rabbit strain”, “mouse strain”, “dog strain”, and “human strain”, respectively [5,18]. The recently discovered strain IV (“human strain”) to date has been found in humans, cats, and dogs [21].

*E. cuniculi* presents a direct life cycle with, most frequently, horizontal transmission by ingestion of spores, and with vertical (intrauterine infection) transmission [5]. Spores, measuring 1.5 × 2.5 mm, can be either ingested or inhaled as the infectious stage of *E. cuniculi* from infected rabbit urine, representing the main source of infection. In fact, spores can be excreted by urine until 1 month after infection and, in wide numbers, up to 2 months after infection. Excretion of spores terminates approximately three months after infection stage, with intermittent shedding of a small number of spores by the infected host [22,23]. *E. cuniculi* spores can survive outside the infected host for up to 6 weeks at 72 °F (22 °C) [11] and for several months under humid conditions [24].

In recent years, different serological surveys with several methodologies, types of animals, and sample sizes were performed in different countries. Particularly in Italy, refs. [1,25] showed a prevalence of 67.2% and 59.5% in pet rabbits, respectively. Santaniello et al. [26] and Lonardi et al. [27] reported a seroprevalence of 31.6% and 75.7% in rabbits from several intensive farms, respectively. Maestrini et al. [28] performed a complex study with a total seroprevalence of 70.5% of analyzed sera. Particularly, their study included rabbit sera from intensive (73.7%) and family (50%) farms, zoos (100%), research laboratories (57%), and owned (44%) rabbits.

The encephalitozoonosis caused by *E. cuniculi* may be acute or chronic, and the clinical signs are induced by lesions affecting the central nervous system, kidney, or eye. Generally, clinical disorders of rabbits infected by *E. cuniculi* include predominantly neurologic signs followed by renal damages and ocular disorders, reported as phacoclastic uveitis [1,29]. In each subject, typically only one of the mentioned clinical forms occurs, although different manifestations have been documented at the same time.

Most rabbits with neurologic manifestations present exclusively with vestibular dysfunction [24,25,26] having a sudden onset and including head tilt, nystagmus, ataxia, circling movements, and rotation with respect to the longitudinal axis of the body. As reported by Künzel et al. [11], in some cases vestibular dysfunction in rabbits with clinical encephalitozoonosis may be highlighted only by subtle swaying at rest or by slight ataxia.

In addition, rabbits with neurologic signs of encephalitozoonosis have also shown paresis, seizures, and behavioural changes.

In rabbit encephalitozoonosis, the set of ocular lesions are known as phacoclastic uveitis and includes cataracts, uveitis, and white intraocular masses, usually detected at the level of the anterior lens capsule [30,31,32]. Most frequently, this ocular lesion is unilateral and involves young rabbits, while bilateral lesions have been occasionally documented [33,34,35].

According to the scientific literature, most rabbits with *E. cuniculi*-associated kidney disease show nonspecific signs, such as polyuria, polydipsia, dehydration, inappetence, and consequent weight loss [29]. The diagnosis, in most cases, is determined through the urea and creatinine dosage, and sometimes, renal damage represents only an occasional finding under blood analysis in adult or older individuals [11].

#### 3.1.2. Rodents

In the past, *E. cuniculi* has been diagnosed in different studies as a common parasite in laboratory animals such as mouse, rat, hamster, and guinea pig [3,5,36], but currently, these infections should no longer be a significant problem in these animal species due to the high applied hygienic standards for the management of laboratory animals. Additionally, rodent models have obtained significant attention for immunological research in the field of microsporidiosis [5,37]. Recently, Kotková et al. [18] provided evidence regarding transplacental transmission of Encephalitozoon cuniculi in mice, although the mechanism is again unclear.

In the past, three reports on *E. cuniculi* infections in wild rats from the United Kingdom and Japan have been published [36]. Muller-Doblies et al. [38] isolated *E. cuniculi* “mouse strain” (strain II) from one of 30 sampled wild rats (*Rattus norvegicus*) in the city of Zurich, Switzerland. Specific antibodies against *E. cuniculi* spores were shown in Iceland in 4% and 9% of wild mice (i.e., *Apodemus sylvaticus* and *Mus musculus*, respectively) [39]. The authors suggested that mice are a potential “reservoir” of *E. cuniculi* for arctic foxes and feral minks, as their favourite prey. In this regard, the molecular characterization of *E. cuniculi* isolates from Finland and Norway showed that foxes originating from four different farms were infected with the “mouse strain” (strain II) of *E. cuniculi* [36,40]. Recently, Meredith et al. [41], identified the *E. cuniculi* seroprevalence in wild rodents, foxes and cats in the United Kindom, highlighting that, in particular, foxes could act as sentinels for the presence of *E. cuniculi* in rodents. Hofmannová et al. [42] described a fatal case of encephalitozoonosis in a group of Steppe lemmings (*Lagurus lagurus*). The animals showed aggression, progressive weight loss, purulent conjunctivitis, cannibalism, and hind limb paresis. Kitz et al. [43] described a case of encephalitozoonosis in a group of captive Barbary striped grass mice (*Lemniscomys barbarus*) in a zoo collection. Perec-Matysiak et al. [44] reported the presence of this microsporidium in wild living rodents from Poland, the Czech Republic and Slovakia, identifying as the most frequent species *E. cuniculi* genotype II (92.5%), followed by *E. cuniculi* genotypes I (1.5%) and III (6.0%).

#### 3.1.3. Cats

As reported by Pang & Shadduck [45], the sensitivity of the cat (*Felis catus*) to *E. cuniculi* infection has been demonstrated during an experimental infection of feline leukemia virus-infected kittens. Transmission is likely to occur through the ingestion of water or food contaminated with infected spores [46]. Unlike what has been reported in rabbits and rodents, it is not clear whether in utero transmission may represent a route of transmission [33]. However, a case of generalized encephalitozoonosis in a kitten with cerebellar hypoplasia could have been related to an acquired in utero infection [47].

Various serological investigations with different methodologies and sample sizes were carried out in different countries, with a serological status between 0% and 26.8% [41,48,49,50,51].

The main clinical signs observed in cats in *E. cuniculi* infections are represented by ocular lesions, while other clinical forms characterized by kidney damage and meningoencephalitis are under discussion [51,52]. In fact, recent studies have not shown a correlation with meningoencephalitis in cats [53] nor in chronic kidney disease in cats [49] in Austria and in Virginia, respectively. On the other hand, various reports have evidently identified *E. cuniculi* in different ocular infections [4,33,54]. Particularly, Csokai et al. [54] identified *E. cuniculi* strain II via PCR in lens material from a pet cat with a cataract and bilateral anterior uveitis. Similarly, Benz et al. [33] showed the presence of *E. cuniculi* DNA in lens and aqueous samples, as well as antibody seropositive status, in 19 cats with cataracts.

#### 3.1.4. Dogs

Among the first reports to attest to the zoonotic potential of *E. cuniculi* by dogs (*Canis lupus familiaris*) are Snowden et al. [19], who identified 8 new isolates of *E. cuniculi* strain III from immunocompromised dogs and humans, suggesting the zoonotic potential of this species of microsporidia. This strain is associated with clinical disease in dogs, but some dogs can be asymptomatic carriers and excrete spores in their urine [19]. Clinical signs in naturally infected dogs include encephalitis, vasculitis, blindness, renal disease, and high mortality [55,56,57,58]. These clinical forms are associated with transplacental transmission of *E. cuniculi* in dogs; however, the infection usually remains subclinical if acquired after birth [59,60]. Recently, de Boer et al. [61] evaluated the presence of *E. cuniculi* in a group of 16 young dogs (<3 years) with neurological signs, showing this parasite is most likely of limited clinical significance in young dogs.

Lindsay et al. [62], in their bicentric study in Brazil and Colombia, showed a prevalence of 14.3% and 35.3%, respectively. Sasaki et al. [63] carried out a serological survey in Japan with a prevalence of 21.8%, highlighting the potential zoonotic risk of this microsporidium. Cray and Rivas [64] reported an antibody seropositive status of 21.6% in a total of 125 sampled dogs. Duzlu et al. [65] evaluated the prevalence of microsporidian parasites, identifying *E. cuniculi* in six (2.1%) samples.

#### 3.1.5. Horses and Donkeys

The equine microsporidiosis caused by *E. cuniculi* is known to cause abortion and placentitis [66,67,68], but microsporidia in horses present a limited scientific interest. Few recent studies have described the seroprevalence and the course of infection of *E. cuniculi* in horses from different countries [69,70,71].

As reported by Laatamna et al. [72], *E. cuniculi* was identified in 1.8% (4/219) and 1.6% (2/124) of horses and donkeys, respectively. In this study, three genotypes of *E. cuniculi* were detected. In detail, I and III were identified in horses; in contrast, only *E. cuniculi* genotype II was detected in donkeys. Particularly, *E. cuniculi* genotype I was identified on suburban farms focused exclusively on horse breeding, while genotypes II and III were detected in rural areas. Cray et al. [73] have conducted a survey to evaluate the seroprevalence of *E. cuniculi* antibodies in sampled horses in the USA. Five or 4.8% of the 105 sampled horses showed seropositivity. One of the samples was taken from a healthy horse. Clinical signs and diagnoses of the other animals included colic, lameness, osteochondritis dissecans, and fever. Wagnerová et al. [70] showed the overall prevalence of *E. cuniculi* in sampled horses was 6.9% (26/377), and particularly, in subjects over 3 years of age, prevalence was significantly higher (10.0%) compared to younger subjects (4.0%). Differently, no significant differences in prevalence were reported among stallions, geldings, and mares, but differences were found in the distribution of *E. cuniculi* genotypes in horses. In fact, genotype I was predominant in housed horses, while genotype II predominated in grazing horses.

Although the relationship between *E. cuniculi* infection in horses and disease remains unclear, the examined studies have shown that horses and donkeys can be parasitized by various *E. cuniculi* genotypes and represent a potential source of infection for humans.

### 3.2. Infection in Humans

The most common microsporidial infections in humans are due to Encephalitozoon (*E. cuniculi*, *E. intestinalis*, and *E. hellem*) and *Enterocytozoon bieneusi* [8]. In particular, as described above, *E. cuniculi* can infect numerous domestic animal species, including rabbits, dogs, and cats, which may serve as possible reservoirs for human contamination. Indeed, these spores are excreted in the urine, faeces, or sputum and are very resistant in the environment. Therefore, ingestion of contaminated food or water are possible sources of human infection [74]. Another route of infection, although less frequent, is inhalation of spores.

*E. cuniculi* is considered an opportunistic pathogen and infection can last more than 1 year. Immunosuppressed people, such as AIDS patients or organ transplant recipients who are being treated with immunosuppressive drugs, are the main target of these spores [75]. In these individuals, spore cells, disseminated in the blood, can migrate to the brain and other sites, causing various lesions in the nervous and respiratory systems, digestive tract, liver, peritoneum, bladder, and kidney. On the other hand, *E. cuniculi* has been also reported in subjects with non-compromised immune systems, such as asymptomatic subjects or travelers returning from tropical countries [9,74,76,77,78].

Individuals who are infected may develop fever, headache, nausea, vomiting, diarrhoea, breathlessness, respiratory symptoms, and weakness. Nevertheless, symptomatology differs according to the age of the person and the concomitant pathologies.

The first human infection with *E. cuniculi* was described in 1959 by Matsubayashi et al. [76], but many other cases have been reported over the following years and some of them will be discussed in this paragraph.

In 1987, Terada showed a case of a 35-year-old man affected by HIV and Kaposi sarcoma presenting with weight loss, general fatigue and occasionally diarrhoea. After 4 months, this patient had a rapid worsening of hepatocellular necrosis due to *E. cuniculi* and died [79]. One year after, Zender [80] reported a case of *E. cuniculi*-induced peritonitis in a patient with AIDS, which was one of the contributing causes of his premature death. Interestingly, in these studies, carried out in the 1980s, the authors did not report a specific treatment of encephalitozoonosis. In 1995, De Groote indicated that in a HIV-positive patient with disseminated *E. cuniculi* presenting with ulcers in the tongue, decreased visual acuity and maxillary sinusitis, treatment with albendazole caused a rapid resolution of the cough and sinusitis as well as disappearance of microsporidia spores from nasal and conjunctival specimens [81]. The efficacy of albendazole was also suggested by Fournier, who described that 5 of 8 immunosuppressed patients reported in the literature treated with albendazole showed a clinical improvement and clearance of *E. cuniculi*. Moreover, in the same article, the author reported a clinical case in which a HIV-positive subject with *E. cuniculi* infection died a few months after hospitalization, after denying treatment with albendazole. A similar case was reported by Fournier in which a 45-year-old man with AIDS presenting with keratitis, bilateral papilledema, renal failure, cerebral lesions and disseminated microsporidiosis denied albendazole treatment and died a few weeks after being hospitalized.

Although albendazole remains the preferred drug in humans with encephalitozoonosis [82], a contrasting result about the effect of this medicine was described by Weber in 1997 [83]. The author reported a case of a 29-year-old HIV-positive patient with *E. cuniculi* infection due to a previous exposition to animals, including rabbits. In this subject, who had multiple cerebral lesions, treatment with albendazole was only successful in the beginning, and persistence of the spores in urine was the probable cause of his death.

Very recent studies indicate an increased risk of *E. cuniculi* infection not only in AIDS patients. In particular, Kicia et al. [84] show that among 72 immunosuppressed renal transplant recipients, 6 were *E. cuniculi*-positive and 5 of them suffered from respiratory symptoms. The presence of *E. cuniculi* was also identified in 1 of 105 patients affected by various respiratory diseases. The authors pointed out the importance of screening for microsporidia not only for the HIV-positive subjects, but also for patients with respiratory symptoms.

## 4. Serological and Molecular Diagnosis

Diagnosis of *E. cuniculi* infections in wild and pet animals is actually based on different strategies, including histopathology, serological testing, such as enzyme-linked immunosorbent assay (ELISA), Western blot, indirect immune fluorescence antibody test (IFAT) and molecular testing, which includes different types of polymerase chain reactions (PCR) [10,50,51,54,85,86,87]. However, *E. cuniculi* diagnosis has equally gained an increase in importance in human health, since the microsporidia can live in both immunocompetent and immunocompromised individuals and cause important illness in the latter [77,80,83,88,89].

### 4.1. Histopathology

Morphological lesions in kidneys and brain can be considered diagnostic for confirmation of encephalitozoonosis in rabbits. Indeed, histopathological diagnosis of granulomatous encephalitis and chronic interstitial nephritis have been corroborated by PCR in more than 80% of rabbits infected with *E. cuniculi*. Although brain, liver and eyes are the most examined organs, focal granulomatous infiltrates have been also observed by histopathology in the lungs, livers and hearts of infected rabbits [54,85]. Interestingly, while PCR has proved especially useful for the diagnosis of phacoclastic uveitis in living *E. cuniculi*-infected rabbits, histological examination combined with special staining has been suggested as the most sensitive method in post-mortem diagnostics [54]. Staining and electron or light microscopy have been used to diagnose *E. cuniculi*-related diseases in humans, as well. In particular, staining techniques such as haematoxylin and eosin, Gram’s, Giemsa, acid-fast and periodic acid–Schiff were used on tissues and samples such as cerebrospinal fluid, sputum, and urine previous to microscopic observation. Spores were detected through their structural characteristics, such as size, shape, dispersion pattern and polar filaments [80,83,89]. Furthermore, concentrated urine was also indicated to be an adequate sample for cytological microsporidia detection both in symptomatic and asymptomatic subjects by Schwartz et al. [90]. Alternatively, trichrome, fluorochrome and fungifluor stains were suggested to be other valuable techniques for common clinical diagnostics in urine, stool and mucus and tissues section samples. Unfortunately, histopathology shows some limits. First, this technique is strongly dependent on clinical signs that positive hosts do not necessarily show. In addition, most of the lesions are not exclusive to *E. cuniculi* infection, and it is therefore necessary to exclude other diseases which might originate similar symptoms [11]. Third, being preferentially performed in post-mortem, histopathology is particularly useful for assessing zoonotic effects but is of poor relevance for clinical diagnosis and treatment in individuals. Finally, some histopathology techniques are expensive, time consuming and can lack accuracy; thus, it is always required to include confirmation or complementization with other diagnosis methods, such as ELISA, Western blot, or PCR.

### 4.2. Serological Testings

ELISA is a serological technique used to detect the presence of antibodies against *E. cuniculi* in the host’s serum, as it is a rapid and well-established sensitive method [91]. However, seroprevalence studies done in Japan and in the USA have shown a high discrepancy. In a study of 163 population of pet cats in Japan, 4.3% had *E. cuniculi* antibodies [50], while 26.8% were found positive in the States in a total population of 127 pet cats [51]. The authors believe the discrepancies are consequences of different geography and/or serological methods. Nonetheless, more studies are needed in order to confirm those hypotheses and infer the reproducibility and accuracy of ELISA as an *E. cuniculi* diagnosis method. Although ELISA is frequently used in serologic Encephalitozoon diagnosis in humans, IFAT has been indicated to be potentially more specific at the genus level [88]. Consequently, IFAT was chosen as the gold standard technique in a study by Abu-Akkada et al. [9] for *E. cuniculi* seroprevalence among 44 non-HIV immunocompromised (group I) and 44 immunocompetent (group II) subjects in Egypt. The results revealed a seroprevalence of *E. cuniculi* of 77.3% and 11.4% in group I and group II, respectively. Samples were next evaluated by microscopic spore examination of Weber’s green modified trichrome stain (MTS) of smears. Compared to IFAT, MTS yielded 68.2% (group I) and 4.5% (group II) positivity rates for *E. cuniculi* spore detection, which translates to a sensitivity and specificity of 69.23% and 89.80%, respectively. PCR was used as another corroboration method using urine samples, but did not gave any positives [9].

Quantitative Western blot was applied to detect antibodies against *E. cuniculi* in rabbit blood samples. Compared to methods like ELISA and many immunofluorescence assays, Western blot has the advantage of not only possessing a higher sensitivity, but also specifically detecting anti-*E. cuniculi* immunoglobulin G (IgG; 135 and 50 kDa) and immunoglobulin M (IgM; 50-kDa), thus allowing for a distinction between past and active infections. Some experiments performed in rabbits led to an estimated sensitivity and specificity of 88.4 and 98.8% for anti-*E. cuniculi* IgG and 84.3 and 98.8% for IgM antibody detection, respectively. Nevertheless, the authors concluded that, due to its labour-intensive and time-consuming features, quantitative Western blot could have a low potential for routine diagnosis of *E. cuniculi* [87].

### 4.3. Polymerase Chain Reaction (PCR)

Different PCR techniques, including conventional, nested, or real-time PCR, are widely used for *E. cuniculi* detection. In rabbits with phacoclastic uveitis, conventional PCR has been considered the best technique to detect *E. cuniculi* [54]. However, in some cases, nested PCR detection in rabbits and humans yielded better results than conventional PCR [8,54]. Despite its high sensitivity, nested PCR does not allow parasitic quantification and is vulnerable to more contaminations [18]. Interestingly, compared to the precedent PCR techniques, real-time PCR is able to also quantify the number of spores in the host’s organism. Experiments in mice using real-time PCR detected a level of parasitic DNA as low as 8 × 10^4^ [18]. In another research, the presence of *E. cuniculi* was tested by real-time PCR in pet rabbits exhibiting neurological symptoms. DNA was collected from urine, and 26.21% (27/103) of the diseased animals tested positive for the presence of the parasite’s genetic material [86]. Although PCR techniques are often used for *E. cuniculi* detection, post-mortem detection gives better results when compared to fluid testing, as both symptomatic and asymptomatic individuals can sporadically shed spores and both positive and negative detection do not guaranty a correct diagnosis [10,18].

In summary, *E. cuniculi* infections represent a great health threat for both human and animals. Thus, a rapid, reproducible, and accurate diagnosis method is needed not only for treatment of this infection but also for zoonotic and seroprevalence studies.

As all the mentioned techniques are still dependent on factors such as individual differences or pathogen distribution and shedding in hosts, the diagnosis of *E. cuniculi*-related diseases may not converge in a single eligible method, as also reported in Table 1.

## 5. Animal Assisted Interventions and Involved Animal Species

As reported by the International Association of Human-Animal Interaction Organizations (IAHAIO), animal-assisted interventions (AAIs) represent different interventions that incorporate the human-animal relationship and teams in formal human services, referred to as animal-assisted therapy (AAT), animal-assisted education (AAE), animal-assisted activities (AAA), and animal-assisted coaching (AAC) [92]. In addition, according to the “One Health–One Medicine” approach, these interventions actively promote collaboration and communication between different disciplines and professionals to work together at local, national, and global levels, via an integrated approach for health and well-being of humans [93,94,95,96].

In recent years there has been a large international scientific production at various levels in the field of AAIs, but most scholars have been interested in AATs, which represent a valid non-pharmacological therapeutic approach for patients with depression, autism, dementia, mental or physical distress, or other illnesses [97,98,99,100,101,102,103,104].

As reported by the IAHAIO [92] (page 8), “Only domesticated animals can be involved in interventions and activities. Domesticated animals (e.g., dogs, cats, horses, farm animals, guinea pigs, rats, fish, birds) are those animals that have been adapted for social interactions with humans. (…) Wild and exotic species (e.g., dolphins, elephants, capuchin monkeys, prairie dogs, arthropods, reptiles), even tame ones, cannot be involved in interactions. The reasons are many and include high risks to clients from zoonoses and animal welfare issues. (…) Not all animals, including many that would be considered “good pets” by their owners, are good candidates for AAI.” In general, regarding involved animals in AAIs, dogs are the most studied animals [14,15,95,101,104,105], but other species are often considered, such as horses [106,107,108,109], cats [110,111,112], rabbits [113,114,115], guinea pigs [116,117,118,119], and cage birds [120].

From the scientific literature screened, most of the studies concern the effectiveness of AAIs, while few scholars have turned their attention to the impact of zoonoses and the related risks caused by contact with animals involved in this type of therapeutic intervention.

In particular, since the dog is the main involved animal species in this type of intervention [95,101,104], most of the studies concerning the zoonoses carried by the animals involved in AAIs, even if small, mainly concern the dog [103,121,122,123,124,125,126]. With regard to zoonoses carried by other animal species in the context of AAIs in general, only Simonato G. et al. [127] has paid attention to the parasites carried by other animal species other than dogs (equids, cats, birds, rabbits, rodents, and goats), while other authors have focused on generic hygiene measures to prevent and reduce zoonotic risk [128,129,130].

Usually, as reported above, AAIs carried out in healthcare facilities are aimed at patients with mental or physical distress, depression, dementia, autism, or other illnesses [97,100,101]. During AAI, patients (often children, young, adult, old, or immunocompromised people) interact with dogs or other animal species through different activities of interspecific types (i.e., petting, leading walking, brushing, hugging, etc.) [101]. In this regard, Shen et al. [131] reported that “bodily contact” significantly influenced the effectiveness of these therapeutic interventions. On the other hand, it is noteworthy that, due to repeated contact with the animal’s mucosae (e.g., mouth, ocular, nasal) and fur (e.g., perineal and genital areas, limbs and tail), patients can be exposed to zoonotic pathogens (e.g., parasites, bacteria, viruses, and fungi) transmitted through direct contact with the animal or with its body fluids or secretions, and by indirect contact with contaminated bedding, water, food, or bites from an arthropod vector [96,132,133]. In order to assess the zoonotic risk of transmission from animals involved in AAIs, the frequency and type of contact between animals and involved people in these therapeutic interventions should, importantly, also be evaluated [96,134].

In particular, AAIs, in addition to being carried out in different ways depending on the animal species, take place in often very different contexts. For example, therapies assisted by equines are carried out outdoors, usually in the context frequented daily by the animal. Differently, for other domestic species, such as dogs, rabbits as well as guinea pigs or birds, the animals are carried in the context where the people involved in AAIs (i.e., hospitals, health facilities).

## 6. Discussion

Zoonoses represent a major public health problem around the world and their prevalence is increasing at an alarming rate. Zoonotic diseases are spread between animals and humans and can be caused by microorganisms such as bacteria, viruses, fungi, and parasites [135,136]. Among the microorganisms that induce opportunistic infections, *E. cuniculi* is a mammalian microsporidial pathogen with a world-wide distribution. This intracellular parasite can infect various animal species, including humans, and is responsible for a disease named encephalitozoonosis [3,8,74,84]. *E. cuniculi* produces environmentally resistant spores that inoculate the infective sporoplasm into the host cell through a polar tube. The release of the spores into the environment occurs following the destruction of the host cell [3].

Acute infection mainly involves the lungs, intestines, and placenta, while in a second chronic phase, it affects the central nervous system and the kidneys. Typical lesions associated with encephalitozoonosis are granulomatous meningoencephalitis, chronic granulomatous, interstitial nephritis and fococlastic uveitis [137,138].

Regarding humans, patients affected by AIDS have been considered to be at greatest risk due to severely compromised immune systems. However, this infection has also been spreading in transplant recipients and people returning from exotic travels. Recently, among the animal species developing encephalitozoonosis, there have been farm animals or even pets, such as rabbits, dogs, cats, horses, donkeys, and humans. In particular, operators of the rabbit meat production chain, such as breeders, health workers (company veterinarians or public officials) or slaughterhouse personnel are certainly among the subjects most at risk. In addition, due to their ethogram and their different history of co-evolution with humans [101], the animal species mentioned above usually are involved in AAIs [104,139]. In this context, interaction between animals and patients can usually include different relational activities such as petting, physical contact, brushing, playing, observation, and strolling with the animal. Particularly, it is very noteworthy that “bodily contact” represents one of the main aspects contributing to AAI effectiveness, even in several settings [131]. In fact, during these therapeutic interventions, patients (e.g., immunocompromised individuals, elderly, and children) are frequently in bodily contact with the animal during AAI intervention, and can thus be potentially exposed to zoonotic agents such as bacteria, fungi, and parasites [96,132,133] when animals are asymptomatic [125]. In any case, regardless of the animal species involved and the mode of interaction/relationship, contact with the co-therapist animal occurs and must occur constantly in all AAI settings, even though this action can expose patients to the risk of zoonosis, especially when specific health checks are not carried out and the hygiene measures recommended by the lacking scientific literature are not applied [128,129,130].

A final category at risk of contagion involves all the workers in the field of research (animal facility workers, researchers, etc.). Thus, an early and accurate diagnosis of infectious disease is of critical importance, since it can improve the effectiveness of treatments, avoid long-term complications for the infected patients and prevent undiagnosed subjects from unknowingly transmitting the disease to others [77,78,80,83,88,89].

Several efficacious and economically viable strategies to diagnose encephalitozoonosis, including histopathology, serological testing, Western blot, IFAT or PCR, are currently available [10,50,51,54,85,86,87].

The diagnosis of encephalitozoonosis can have several difficulties due to various problems, including the poor sensitivity and specificity of the diagnostic tests or the scarce evidence of clinical signs and symptoms. Currently, laboratory tests are fundamental in the diagnosis of encephalitozoonosis. Microscopic examination can give useful indications for therapeutic treatment and, in some cases, allow a presumptive identification of *E. cuniculi*. However, the sensitivity of microscopic examination depends on the type of sample to be analyzed, the amount of material available and also the choice of operating procedures for the microscopic preparation. At the same time, serological tests, despite having a higher sensitivity and specificity, are conditioned by the physical condition of the patient. Indeed, most subjects at risk of encephalitozoonosis are immunocompromised, and the production of antibodies can be reduced. Furthermore, the production of antibodies, due to the specific immune response, takes a few weeks to reach appreciable levels, which can be extremely limiting. Different molecular testing options, including conventional standard PCR, nested PCR, and real-time PCR, are available for the qualitative and quantitative detection of *E. cuniculi* and for its identification. These methods provide information and show a wide range of performance in terms of analytical sensitivity and specificity. However, even in this case, the clinical interpretation of PCR-positive samples may not be easy and may require rather lengthy procedures. Data about specific therapies are also still unclear. Albendazole remains the most effective drug, although a specific plan of direct prophylaxis should be adopted to reduce infectivity [75]. However, the lack of information in the epidemiological field and the difficulty of making a diagnosis do not allow an adequate application of prophylactic measures and therapies.

It would be desirable to develop new, more precise methods to highlight the infection induced by *E. cuniculi* in order to allow for more concrete control of the spread of this pathology and better care in humans and in animals.

## 7. Conclusions

In recent years, emergency episodes due to new zoonoses have occurred with increasing frequency. There are many reasons for the increasing zoonotic disease emergencies, and several elements contribute to it, including changes in the climate or ecosystems, human or animal demographic changes, urbanization, close contact between humans and animals and changes in pathogens caused by mutations, natural selection, or evolution [140].

Encephalitozoonosis represents a concrete pathological risk not only for humans, but also for animals in contact with them, which must be prevented and treated.

Therefore, it would be advisable not only to deepen the studies concerning the potential pathogens transmitted by the animals involved in AAIs but also to create a network of connection between scholars to standardize health protocols useful for fully protecting patients according to the context and the animal species involved.

## Figures and Tables

**Table 1 ijerph-18-09333-t001:** Summary table of *E.cuniculi* diagnostic methods.

Diagnostic Procedures		Highlights	References
Serological tests	ELISA	Rapid Well-established	[50,51]
	Western Blot	Allows specific detection of IgG and IgM (qWB)	[4,87]
	IFAT	Specificity to the pathogen’s genus	[9,88]
Molecular tests	Conventional PCR	High sensitivity using samples from phacoclastic uveitis material	[54]
	Nested PCR	Rapid High sensitivity in all organs and body fluids	[8,54]
	Real-time PCR	Rapid Quantification of the spores	[18,85]

## Data Availability

Data are available upon request to the corresponding author.

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
