# Peer review of "Zoonotic Risk of Encephalitozoon cuniculi in Animal-Assisted Interventions: Laboratory Strategies for the Diagnosis of Infections in Humans and Animals"

_ijerph, 2021, doi:10.3390/ijerph18179333_

Round 1
Reviewer 1 Report
The article is devoted to a rather urgent problem of the safety of the use of zootherapy and coaching with animals. The authors have compiled a detailed review, including methods for diagnosing the microsporidiosis under consideration, pathogenesis and other aspects of diseases. Based on information from literature sources, it is shown that in patients with weakened immunity, this conditionally pathogenic parasite can cause severe symptoms. At the same time, the article weakly shows the relationship between the risk of infection during animal therapy, and not with other types of contact with animals, in which basic hygiene measures were not observed. The authors ' statement that there is currently an increase in the incidence of zoonotic infections requires confirmation by literary data. The mention of the spread of the COVID-19 virus does not seem entirely appropriate, since the virus is currently being transmitted from person to person, and this is why a huge increase in the number of cases is associated with this. In addition, it is not entirely clear from the literature sources given in the article whether the premature death of patients with immunodeficiency was associated with the presence of only this microsporidium or there was a combined infection.
Reviewer 2 Report
ijerph-1341846
Zoonotic Risk Of Encephalitozoon Cuniculi In Animal Assisted Interventions: Laboratory Strategies For The Diagnosis Of Infections In Humans And Animals
Antonio Santaniello Ilaria Cimmino , Ludovico Dipineto , Ayewa Lawoe Agognon , Francesco Beguinot , Pietro Formisano , Alessandro Fioretti , Lucia Francesca Menna , Francesco Oriente
General: This is a well-written review on an important topic that most people in the fields of One Health and AAI tend to ignore. As such it deserves significant attention. It is a good summary and complementary introduction to the topic for groups from both of these fields. It is sufficiently detailed and documented to be helpful but also is very concise.
I have made a few suggested edits that the authors can choose to accept.
Specific: I have suggested some word changes at various places in the paper as indicated by the ‘notes’
Page 3: first paragraph. I am not certain what the authors want to communicate here.
Page 6: the authors often include the first initial of the author they are referencing. I would suggest deleting the first initial unless this is the journal format.
Page 6: in much of the paper the authors italicize E. cuniculi but on this page they often do not do that. Is there a reason for that?
Page 8 line 342: can they identify what (sized?) proteins are being detected in Western blot?
Other minor suggested edits follow.
